# Professional Training of Employees in Media Organizations in Serbia and Its Implications on Career Development

**Bojana Sokolović, Iva Šiđanin * , Ljubica Duđak *  and Sonja Kokotović**

Faculty of Technical Sciences, University of Novi Sad, Trg Dostiteja Obradovica 6, 21000 Novi Sad, Serbia
* Correspondence: iva.sidjanin@uns.ac.rs (I.Š.); ljuba@uns.ac.rs (L.D.)

**Abstract:** This paper examines the existence of the need and willingness of media companies in Serbia to organize professional training for their employees, with the aim of improving their knowledge and skills and for the sake of their career growth, analyzed in the context of business sustainability, as well as the sustainability of human resources in a very competitive media industry. The main purpose of the paper is to determine the relationship between professional training for media professionals and their career growth. It looks into the subjective assessment of the need for training, the subjective assessment of the usefulness of training, and organizational support related to training and development in order for media professionals to achieve career growth. A correlation study was conducted on a sample of 150 media professionals using three scales of a job training and job satisfaction survey questionnaire to determine the subjective assessment of the need for professional training, the subjective assessment of the usefulness of professional training and organizational support to training, while a modified organizational career growth scales questionnaire was used to determine career growth. Research results confirmed that employees believe that additional training has an impact on the development of their career, as well as that attending additional training has a positive effect on their career growth. This paper provides useful guidelines for company management regarding professional training and its implications on career growth, theoretical and practical implications, as well as suggestions for future research.

**Keywords:** training; journalists; career development; organization; sustainable development

## 1. Introduction

Career growth, which is connected to employee training and development, is one of the key assignments of every modern organization. The contemporary age brings about different demands regarding the sustainability of human resources, including media professionals, since they are employed by media companies. The technological revolution, digitalization, and globalization of society point toward the significance of educating employees in the media industry. This also includes media professionals in Serbia who need to be prepared for future challenges, taking into account the country's accession to the European Union. Here, a special emphasis has been put on opening the chapters which regard the media. Regarding this, it is necessary to research how media companies approach the professional training of media professionals and in what way such training may influence their career growth.

The lack of sufficient similar research in the field of employee professional training and development and their relationship to career growth conducted within the media industry sector in Serbia has led to the creation of this paper. In research papers that deal with human resources [1], there were mentions of career management, and this is directly connected to the professional training of employees. However, in Serbia, this has not yet been confirmed when it comes to media companies. For this reason, it is necessary that this gap between theory and practice be filled, and the following study aims to achieve this goal.

Free and independent media are one of the basic backbones of a democratic society, the driver of democratic transition, and one of the requirements of democratic stability. In order for the media to function properly, the responsibility and professionalism of journalists are necessary, and this requires their continuous professional and personal development, as well as frequent additional training. The work and activity of journalists today are full of challenges, both political and ideological, as well as those brought by the digital transformation of the mass media, which is often said to have ushered journalism into a "golden era", on one hand, and on the other hand, accelerated the problems in the media industry in terms of the decline of traditional media and the mass unemployment of journalists.

Although the common saying that a journalist "must be born" is still often heard, it is common knowledge that for anyone who pursues journalism as a career, it is no longer enough to be gifted and to be able to write well. Journalism practice used to be acquired in newspaper editorial offices where the intern journalist worked under the watchful eye of the editor and with the help of advice from his senior colleagues. From a technical point of view, this is still a good method, but it is not sufficient.

For this reason, the preparation of journalists for present and future business is achieved through knowledge and skills upgrades, by increasing the abilities that are achieved through training within the organization [2]. Although there is now a widespread and growing belief that specialized training for journalists is inevitable, it is not yet universal and it will not be amiss to re-examine the assumptions on which such a belief is based.

De Hauw and Greenhaus [3] draw attention to the fact that several economic, organizational, and social trends that prevail in the working environment today, they additionally emphasize the need for employees to adapt to new changes in order to build sustainable careers thanks to which they remain "employable, healthy and happy over the long term" (p. 223). One of the most important elements of sustainable careers is the training provided by the organization. Such training has an effect on work performance and employability, which is one of the basic elements of the sustainability of employees in an organization.

Media industry employees are well aware of the opportunities modern media business offers, or a lack thereof, as well as of the imperative for media companies to change the ways in which they reach the audience, and of the great competition in the media market, the diversification of the format of journalistic announcements (stories), etc.

In their conclusions and recommendations, the Unesco secretariat [4] especially emphasizes the importance of knowing, for all of those who plan to work in media companies, "what to expect of it and what the profession expects of them" (p. 15). All the efforts that are invested in the process of personnel selection, their additional education and adequate compensation for the work performed, are of great benefit to improving the work of media professionals.

After analyzing the available data related to scientific research in the field of training and career growth of the employees from media companies in Serbia, the authors of the paper concluded that there is a very modest database, which contributed to the decision to study this topic in more detail. Therefore, in this paper, the focus is put on investigating the existence of the need and willingness of media companies in Serbia to organize professional training for their employees with the aim of improving their knowledge and skills, and also contributing to their career growth. This is of particular importance nowadays, due to the conditions of turbulent changes in the business environment where unforeseen circumstances employees need to deal with arise every day. It is very important to achieve the sustainability of business and the sustainability of human resources in order to survive in the highly competitive media market.

The goal of this research was to determine whether media companies organize training for their employees and how this is related to their career growth, that is, to determine the degree of connection between professional training and the career growth of employees in media companies. Additionally, the goal was to determine how many employees attended

professional training courses and to what extent attending such courses had a positive effect on their career development, as well as whether attending training courses was reflected in career growth depending on their socio-demographic characteristics.

Through examining the extensive literature and the data available on the education of media professionals in Serbia, the following research question has been defined: Is there a connection between employee training and career growth?

By proving the set hypotheses, the authors want to confirm the interdependence between employee training and career progress. Therefore, this paper analyzes the impact of training on the career progress of media professionals, viewed through training modalities and the subjective assessment of the need for training, the subjective assessment of the usefulness of training, and organizational support for training. Career progress is examined through different elements of career progress (achievement of personal career goals, advancement on the hierarchical ladder, salary increase, etc.). It also examined whether the demographic characteristics of employees have an impact on career progress. By providing proof for the first three hypotheses, the influence of training elements on career progress was confirmed through correlation and regression analysis. The fourth hypothesis indicates that demographic characteristics are not a decisive factor in career advancement, which suggests that training and education are. Organizational support for training was emphasized as a decisive factor for the career advancement of employees in media companies. Furthermore, career progress is largely conditioned by the subjective assessment of the usefulness of professional training. The contribution of this paper is reflected in the recommendations for the management of media companies on how to increase the satisfaction level of both the company and the employees by ensuring that the status and position of the employees in media companies are in accordance with European and international standards. At the end of the paper, the authors analyzed the limitations and gave suggestions for future research.

## 2. Theoretical Background

### 2.1. Importance of Professional Training and Additional Education

On-the-job training should cause a change in the employees' work-related knowledge, attitudes, and behaviors [5], which means that employees change their knowledge and skills related to their work or the relationships that arise from the work they perform. Training refers to formally planned participation in activities aimed at providing employees with additional knowledge, skills, and attitudes that are useful for performing their current job ([6,7] according to [8]). Companies formulate entire programs that are implemented through the realization of annual or semi-annual training plans and are contained in the strategy for the purpose of education and training of their employees. Training is a prominent means of providing learning activities [8] (p. 10). Learning activities are an integral part of all training plans in organizations and represent all those activities that increase the knowledge related to the work of employees. Landy [9] defined job training as "a set of planned activities on the part of an organization to increase the job knowledge and skills or to modify the attitudes and social behavior of its members in ways consistent with the goals of the organization and the requirements of the job" (p. 306). Professional training includes a set of planned activities that are carried out at the workplace by learning through work, additional external training, and mentoring, and they aim at increasing the employee's abilities and skills to effectively contribute to the achievement of company goals. In order for individuals to obtain and keep a job in a changing labor market, they must increase their needs for career competencies that can help them manage their own careers [10].

Greater competence when performing their own work gives individuals the security to remain in the existing company on satisfactory terms, as well as to present themselves more competitively on the labor market, if necessary. Many authors believe that through the acquisition of new and the improvement of the existing knowledge, skills, and abilities, employees become more productive and achieve better organizational results [11–13],

which has a positive effect both on the work and motivation of the employee, and on the company due to the achievement of set organizational goals. New and innovative knowledge of employees is more likely to be generated in those organizations where there is mutual cooperation between them, as well as a willingness to exchange acquired knowledge and experiences [14].

When companies increase investments in employee training (through financial and other types of support), they help them develop their knowledge and skills, but also contribute to improving the efficiency of the organization itself. In this sense, Wesley and Skip [15] stated that the most obvious benefits related to training are consistency in work performance, greater job satisfaction, user satisfaction, as well as the reduction in business costs. Slavic and Berber [16], on the other hand, emphasized that training and development of employees are seen as imperative in improving the intellectual capital of employees, and also the competitiveness of modern organizations. The degree of increase in work performance and employability after training was the result of training provided and supported by the organization and its management [8].

### 2.2. Sustainability of Employees in Media Organizations and Their Careers

Bozionelos et al. [8] confirmed the essential role of investing in employee training in order to promote sustainable careers, considering that training organized by the organization has a great effect on work performance and employability as the basic element of the sustainability of employees in the organization. The results of their research indicated the connection between the training manager's support and the employees' motivation to learn, their expectations from the training, and the transfer of knowledge during learning or training. The indicator of organizational success is the alignment of an individual's career with organizational interests [17].

Nowadays, the term sustainability is being used more and more. In this paper, it is used in the context of the sustainability of human resources in organizations, including employees in media companies. Furthermore, among other things, the elements of the sustainability of human resources and the sustainability of their careers in organizations are discussed. Quite a few authors dealt with the interdependence between career development and sustainability, that is, sustainable careers in organizations [10,18,19]. The psychology of sustainability and sustainable development, which can also be perceived as a benefit that contributes to the development of the company, is aimed at improving interpersonal relations, both in the organization itself and in general in all walks of life, primarily contributing to the sustainable development of each individual [20].

Sustainable careers represent "a series of career experiences reflected through different patterns that repeat over time, thereby passing through several social spaces, characterized by individual agency, and thereby providing meaning for the individual" [21] (p. 5). In order to achieve the sustainability of the careers of its employees, it is necessary to establish the interdependence between the training and education of employees and progress in their careers. In the modern era of business, it is necessary to enable digital literacy, inclusion, and engagement that encourages the expansion of knowledge and the ability of employees to adopt this knowledge [22].

This is a particularly important aspect of managing sustainable careers and sustainable development in organizations. Those that aim to practice sustainable human resource management should involve their employees in the sustainability process through working groups, training and the development and implementation of a sustainability strategy [23], and that implies inclusion in all training programs that are implemented at the level of the entire organization or through external education programs.

### 2.3. Employee Training in Media Organization, Education, and Career Growth

In terms of career development in the field of journalism, it is necessary to point out some specifics. Digital technologies have influenced the emergence of new roles in media organizations, which forced a reconfiguration of the workplace [22] and adapta-

tion to atypical forms of journalistic work. Therefore, a complex transformation of the profession [24] took place, which creates uncertainty in the future of career development, and eventually, in the possibility of keeping a job. Apart from structural changes in the media sphere, increasing saturation of social media contributes to the insecurity of the journalistic profession, which professional journalists increasingly perceive as a threat to their professional reputation and the maintenance and development of their careers. They write very engagingly about this problem [25], stating that the aggressive environment of social media increases the affectivity of the work of journalists, dissatisfaction with their communication environment, and the need to create a new strategy on professional and personal limits of the use of social media in professional journalism.

Other forms of training are short or extended courses for journalists, established under the auspices and management of professional institutions, the so-called schools of journalism, or media professionals employed by various organizations and associations, as well as the media company itself. They often offer valuable basic education and advanced training for those journalists who wish to supplement their pre-professional studies or who feel the need to enrich their educational and technical experience. In the context of journalist training, Ruggiero et al. [26] emphasize the importance of ethics training, which in turbulent media markets is a counterweight to the numerous challenges that journalists face in the professional performance of their work.

Modern courses have significantly enriched their curricula, which now include training for a whole range of journalistic techniques and skills that are necessary for a successful career in journalism. Otherwise, the principles and methods of modern professional training are formed in accordance with certain international standards, established by large news agencies with networks and connections that cover most countries. In addition, specialized training publications produced by professional associations can be found in many countries.

Ballot et al. [27] stated numerous benefits of training which are planned and implemented by organizations: the improvement of work performance in specific jobs, the increase in employable candidates, the possibility of work engagement during a lifetime, the possibility of increasing earnings, and commitment to work, especially drawing the attention to the benefits achieved by young people and trainees.

Furthermore, there is another important dimension regarding the employees, and that is their career development. As the authors stated [21], upgrading knowledge through additional learning and training positively affected the achievement of sustainable careers. Career development can be defined as the increase and expansion of an individual's abilities, which occurs within an organization. Additionally, chances for advancement and development are closely related to the organization to which an individual belongs [28].

In this sense, Weng [29] presented the concept of career growth through four factors: the progress of career goals, the ability to develop professionally, the speed of promotion, and the growth of personal earnings (salary). Organizational career growth is in the service of the employee, in terms of his or her personal efforts toward the progress of personal career goals and the achievement of professional skills in the organization itself, but also in the service of the organization, through various organizational efforts to reward the actions of the employees in the context of their progress, through various benefits, and an increase in earnings [28].

The literature on media organizations emphasizes that traditional ways and practices are not enough to provide media professionals with the opportunity to deal with the amount of information at their disposal, especially due to social media. This indicates the need to use new knowledge and tools on the basis of which to ensure smooth work for media professionals, which implies additional training and on-the-job training [30]. On the other hand, it has become necessary that the training for journalists include different types of interactive courses that enhance team and individual career growth through the development of skills for recognizing video, text, audio, graphics, and other content [31].

Taking into account the need for media professionals to improve through additional training in order to achieve career progress, the first of four hypotheses was proposed:

**H1.** *Training in a media company is related to the career growth of employees.*

As employees' jobs are increasingly automated, access to new, advanced knowledge and skills becomes critical for them and their further career advancement [32].

Accordingly, the increasingly intensive development of robotic or automated journalism cannot be bypassed; now, there is software that is programmed to write simple news reports and relieve journalists of their daily routine tasks. According to a recent report by the British public media service BBC, by 2022, about 90% of all news will be written by "robots" [33]. This presupposes the creation of an automated narrative, created by artificial intelligence software writers, experts in the field of linguistics, and a good general knowledge of language and grammar. At first glance, the automation of media reporting will enable journalists to devote themselves more intensively to investigative journalism, but on the other hand, it will inevitably lead to a reduced number of jobs for the journalistic profession. It is very important that managers in human resources of media organizations deal with the problem of reduced engagement of the workforce and in time reorient employees to other types of journalistic work that AI (artificial intelligence) and automation cannot replace. One the example is the training of journalists for the use of modern technology (drones, use of satellites, etc.).

Journalism education in most countries around the world traditionally covers the field of training practical skills and standards on the one hand, and general contextual education courses on the other [34]. When it comes to the methods and techniques of professional training of journalists, the first place is still training "on the job", without a formal scheme or additional, prescribed education. The learning "by working" method can be described as a traditional method, and it is still very common in the world of newspapers, radio, and television. It is recommended that managers of media companies focus on new, modern ways of doing business and provide their employees with training for new ways of working accordingly.

In his study, Gaunt [35] presented a detailed report on the training of employees in media companies around the world, defining six levels of the journalist training system: (1) orientation, which implies an understanding of the media system in which journalists work; (2) basic skills, such as writing, editing, and good language skills; (3) technical skills, such as the use of technical equipment; (4) improvement of skills, mainly intended for established journalists (especially technical skills); (5) understanding of social, cultural, and economic issues in society; and (6) specialized applications, such as various fields of mass communication which require specialized training.

Employees who received training using the training methodology felt that it was most effective in helping them to learn (and, thereby, the method they most preferred) were significantly more satisfied with that training than employees who preferred a methodology other than the one that was used most often in their training [36] (p. 493).

In their research, Buonomo I. et al. [37] proved that high job training satisfaction is related to positive perceptions of employees toward learning and progress. On-the-job training represents the main competitive advantage and the source of sustainability of human resources in companies. Therefore, company management should pay attention to the employees' perceptions of professional training conducted by the organization and regularly implement training strategies.

An important fact is that the training aimed at media professionals should be adjusted to their needs at work, but also the employees themselves should be interested in their professional progress. Due to this, hypothesis H2 was put forward:

**H2.** *There is a connection between the individual's personal attitude toward additional training and career growth.*

Employees become more attached to the organization if it provides them with training and encourages them to progress in their careers, both in terms of advancement on the hierarchical ladder and in terms of ability enhancement that enables easier and better performance of the work duties, and that is often reflected through an increase in earnings in the particular organization. Organizations that do not include the costs of employee training in their business plans do not give the impression of a company where an individual would like to stay. On the other hand, an individual to whom the organization has paid attention in terms of training and career advancement is motivated and eager to use his or her expanded knowledge through the performance of activities at the workplace and contributes to the effectiveness of the work of the organization in which he or she is employed, thus the organization comes to the item of saving costs. Given that employees' jobs are becoming increasingly automated, new ways of working are being approached, so advanced knowledge and skills are becoming crucial for employees in their further career development [32]. Organizations that provide mechanisms for the career growth of their employees have a comprehensive investment that implies a strong connection with their employees, and that connection further combines career development with significant results such as, for example, commitment to the organization [28].

Bozionelos N. et al. [8] showed in their work that sustainable careers in the organization can be positively influenced by training and courses organized by the employer, as a result of the learning process. They emphasize the support of the supervisor during the training, as well as the role of the parent organization's support.

Nguyen et al. [38] believe that encouraging employees to access and develop knowledge, ideas, and skills motivates the employees themselves and that on-the-job training supported by the organization can greatly contribute to this, and at the same time increases commitment to the organization.

Career progress is significant both from the perspective of the organization and from the perspective of the employee. This paper will try to prove that through hypothesis H3:

**H3.** *There is a relationship between the attitude of the organization toward additional employee training and career growth.*

The media practitioners should establish a professional body for regulating, improving, and ensuring that professionals are ready and equipped to cope with changes in the media and that there is continuous training for employees in order to improve their professional standards. Regular analysis of the needs employees have regarding professional training is necessary. There should be a requirement for all media practitioners to belong to a professional body, without which one should not join the world of work [39].

Hypothesis H4 was developed as an additional hypothesis to determine whether there is a relationship between training and career growth depending on socio-demographic characteristics.

**H4.** *The attitude of the employees toward training regarding their career growth will differ depending on the socio-demographic characteristics of employees (gender, level of education, years of work experience in the media industry, years of total work experience, etc.).*

The main purpose of this research is to determine the relationship between the professional training of media professionals and their career growth. It is necessary to point out if there is a relationship between subjective assessment of the need for training, subjective assessment of the usefulness of training, organizational support for training, and career growth for the media professional (Figure 1).

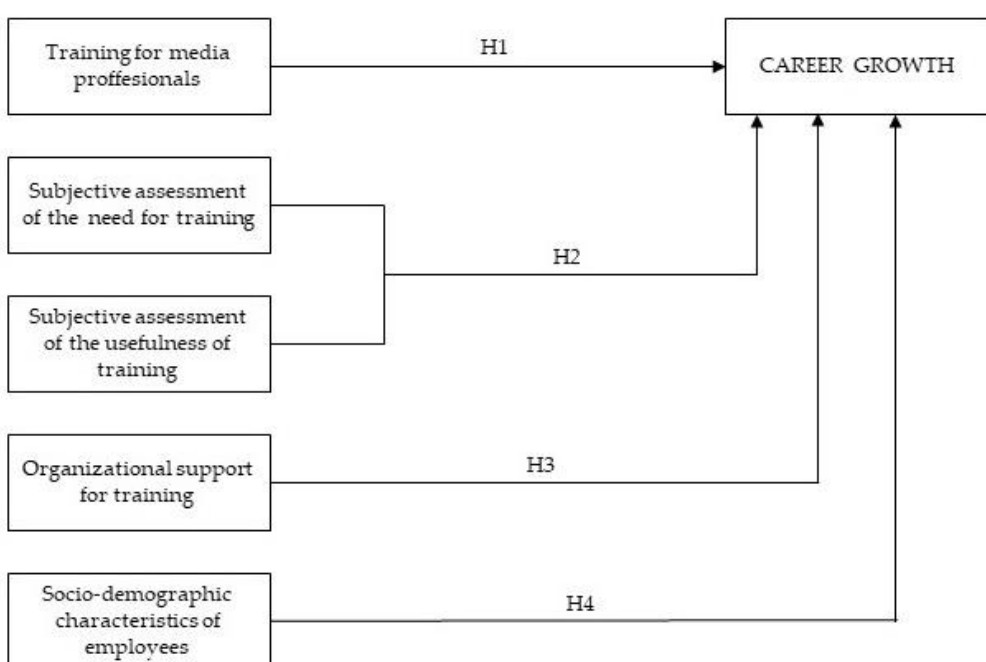

**Figure 1.** Theoretical framework.

*2.4. Professional Training of Employees in Media Organizations in the Republic of Serbia*

When it comes to the Republic of Serbia, there are several associations and organizations that implement training programs for journalists. For example, BIRN Serbia (Balkan Investigative Reporting Network) creates journalist training, seminars, and workshops for representatives of local and national media. They cover various topics, such as reporting on European integration, public finance, local budgets, topics of importance for minority communities, etc. In addition to journalists, civil society, minority communities, local self-government, state administration, etc., are often included in these training courses.

The associations of journalists, UNS (Association of Journalists of Serbia) and NUNS (Independent Association of Journalists of Serbia), are extremely active in the organization of various training courses, such as multimedia journalism courses that enable the acquisition of knowledge and skills required for working in digital media, or specialized training in the field of investigative journalism, transparency of public finance, accountability of public officials, and human rights, with special reference to labor and related rights.

The Association of Journalists of Vojvodina, for example, organizes numerous photography courses, and in 2017, it also organized a three-month video journalism school.

By establishing the Department of Journalism at the Faculty of Philosophy in Novi Sad, the Novi Sad School of Journalism stopped training journalists directly and focused its activities on projects aimed at raising professional capacities in the media sector, monitoring and analyzing media content, and educating different target groups on media literacy, as well as public advocacy of contributions to the democratization and development of civil society and the promotion of European values.

Additionally, many media companies organize both internal and external training for their employees, with the aim of their rapid education that will enable them to compete in the labor market.

The expediency of additional education by attending journalistic "schools", courses, training courses, and lectures is best reflected in the acceptance of their issued diplomas or appropriate certificates on the basis of which the management recognizes their internal qualification.

### 3. Materials and Methods

The sample for this research consists of 150 media professionals employed by television, radio, and newspaper companies, as well as by web pages, newspaper agencies, and media production companies in Serbia, who are, at the same time, members of the Journalists' Association of Vojvodina. The sample was semi-structured since all of the available active members of this media organization, i.e., 53.76%, participated in the study. Participation in the survey was anonymous and voluntary.

The data were collected during February and March 2022, using the printed version of the questionnaire that was distributed to the participants. The questionnaire contained twelve questions, grouped into four units: socio-demographic questions, the attitude of media professionals toward additional training and improvement, the attitude of the media company toward additional training of media professionals, and the attitude of media professionals toward personal career growth.

Some of the questions from the questionnaire were taken from standardized questionnaires, with certain modifications compared to the original and in accordance with the research area and regulations of the Serbian language. There were also additional questions, defined in accordance with the subject and goal of the research. The questions related to determining the attitude of the individual and the organization toward training were taken from a standardized questionnaire by Steven W. Schmidt [36], while the questions related to career growth were taken from Paola Spagnoli and Qingxiong Weng [40].

For the purposes of researching the employees' subjective assessment of the need for training, the usefulness of training, and the organizational support regarding that training, a subscale of the job training and job satisfaction survey with 55 items developed by Steven W. Schmidt was used in order to measure different aspects of workplace training. To measure the elements of career growth, a part of the questions from the OCG scale (organizational career growth scale), also developed by Steven W. Schmidt [36], was used, while the other questions related to career growth were taken from Paola Spagnoli and Qingxiong Weng [40].

Appendix A contains the questionnaire used in the research with the scales adapted to the research goals. The reliability of the questionnaire was checked using Cronbach's alpha. The validation of the scales was performed, where Cronbach's alpha is 0.913 for the entire instrument, i.e., the questionnaire. The subscales were validated as well, so that Cronbach's alpha for the subscale which measures the employees' attitudes toward training is 0.858, and Cronbach's alpha for the subscale which measures career growth is 0.845 (Table 1). In Table 1, it can also be noticed how the entire scale demonstrates very high reliability, while the subscales S1–S7 and S8–S18 also show satisfactory and high reliability.

**Table 1.** The validation of scales and subscales of the research questionnaire.

| Scale | Number of Items | Cronbach's Alpha |
|---|---|---|
| Total S1–S18 | 18 | 0.913 |
| S1–S7 | 7 | 0.845 |
| S8–S10 | 3 | 0.693 |
| S11–S13 | 3 | 0.668 |
| S14–S18 | 5 | 0.891 |
| S8–S18 | 11 | 0.858 |

In the case of socio-demographic questions and one question which refers to the training (question 8), a multiple choice scale was used, while for all other questions, a 5-point Likert scale was used.

The collected data were analyzed using adequate statistical methods. The results were summarized in a table and graphically, while the analysis was performed in IBM SPSS 26 software (IBM SPSS Statistics for Windows, Version 26.0 Armonk, NY: IBM Corp). Numerical variables are described in standard measures of descriptive statistics, while

categorical variables are described by percentages and frequencies. The Mann–Whitney U test and Kruskal–Wallis tests were used to compare the median values of the variables of two independent samples or more than two samples, respectively. Pearson's correlation coefficient was used to test the association between variables. The chi-square test was used to examine the differences in characteristics between categorical variables. $p$-values of <0.05 were considered to indicate statistical significance. Regression analysis was applied to measure the influence of the demographic data on professional training and the influence of the training elements on career growth.

## 4. Results

The results of the conducted research indicated that the gender structure of the research participants was almost evenly distributed—49% were men and 51% were women. Half of the research participants belonged to the age category between 31 and 40 years, while a quarter of the respondents (25%) belonged to the age category between 41 and 50 years. Only 15% of the respondents were over 50 years old and 10% were under 30 years old.

The research participants were almost equally employed in media companies that had national (33%), regional (34%), and local (33%) coverage. More than one-third of the respondents were employed on television (37%), while a quarter of them were employed on internet portals (24%). A total of 15% of the research participants were employed in radio companies, almost as many as in newspaper companies (14%). The smallest number of research participants were employed in media production (7%) and news agencies (3%). A total of 42% of respondents were employed as journalists, 30% as editors, and 17% as associates.

More than half of the research participants (63%) attended more than one training during their career, while 19% of them attended at least one training. A total of 18% of the respondents did not attend any training during their careers.

### 4.1. H1

It was examined whether attending training is related to the following statements about personal professional development, i.e., career growth.

Seven individual statements were given in the questionnaire, which were in agreement with the general hypothesis H1. Six out of seven statements were confirmed, which leads to the conclusion that H1 is also confirmed.

S1: If I attend training according to the set training plan, I will increase the chances of reaching my career goals.

S2: My supervisor encourages me to constantly acquire new knowledge and skills related to the current job.

S3: The training that I would attend, which are foreseen by the training plan, can enable me to progress in the organizational ladder in the next year.

S4: My current job allows me to continuously improve my professional skills.

S5: The probability of advancement in the media company where I am employed is high.

S6: Compared to my colleagues, I was promoted faster.

S7: The knowledge and skills I have acquired through training during my career have contributed to my progress, which is reflected in an increase in salary.

There was a statistically significant difference in statements that referred to career growth based on whether the participants attended training: Kruskal–Wallis H = 11.852, $p$ = 0.003. The employees who have attended professional training graded career growth with higher values than those who have not. This implies that hypothesis H1 is confirmed (Table 2).

**Table 2.** Attending training sessions in correlation with the career growth statements.

|  | Yes, More than One | Yes, One | No |
|---|---|---|---|
| Career growth | 3.00 | 2.86 | 2.29 |

There was a statistically significant difference in the ratings of the above statements depending on whether the respondents attended training (Kruskal–Wallis H = 21.402, $p$ = 0.000; S2: Kruskal–Wallis H = 7.087, $p$ = 0.029; S3: Kruskal–Wallis H = 13.680, $p$ = 0.001; S4: Kruskal–Wallis H = 7.961, $p$ = 0.019; S5: Kruskal–Wallis H = 8.936, $p$ = 0.011; S6: Kruskal–Wallis H = 7.135, $p$ = 0.028; S7: Kruskal–Wallis H = 1.948, $p$ = 0.378).

Research participants who have not attended training courses or attended only one training session rated most of the statements with lower marks compared to the respondents who attended several training sessions during their career (Table A1). Six out of seven statements were confirmed, which also leads to the conclusion that hypothesis H1 is confirmed.

*4.2. H2*

An individual's personal attitude toward additional training and the respondents' motivation for additional training were examined using the following statements:

S8: I always find opportunities for learning and additional training.

S9: I am proactive in looking for ways to improve professionally.

S10: I am ready to learn more in order to achieve my professional goals.

Table A2 shows the results of the correlation analysis. The existence of statistically significant correlations between the statements indicating personal attitude toward additional training and motivation for additional training and the statements indicating career progress was established. Statistically significant, positive correlations were established for almost all statements. Based on this, it is concluded that there is a connection between personal attitude toward additional training and motivation for additional training and career growth. A positive ratio of correlations means that an increase in positive attitudes toward additional training follows greater career progress and vice versa. The established correlations range from low to medium strength.

There is a statistically significant, moderate, positive correlation between career growth and the subjective assessment of the employees about the need for training courses (Table 3).

**Table 3.** The subjective assessment of the need for training correlated with career growth.

|  |  | Subjective Assessment of the Need for Training |
|---|---|---|
|  | Pearson's correlation | 0.294 ** |
| Career growth | Sig. (2-tailed) | 0.000 |
|  | N | 150 |

** Level of significance 0.01.

In Table A3, the attitude toward training is illustrated (defined by the following statements):

S11: The knowledge I acquired during my journalistic education significantly contributed to the quality of the work I perform.

S12: I believe that the knowledge acquired during training sessions in my field is very applicable in my work.

S13: I am very satisfied with the frequency of training organized by the media company where I am employed and the progress in my career.

The obtained results indicate the existence of statistically significant, positive correlations between almost all the examined claims.

There is a statistically significant, median, positive correlation between career growth and the subjective assessment of the employees of the usefulness of training (Table 4).

Based on the obtained results, it has been concluded that hypothesis H2 is confirmed, since the results have shown that there is a connection between the employees' personal attitudes toward professional training and their career growth.

**Table 4.** The employees' subjective assessment of the usefulness of training.

| | | Subjective Assessment of the Usefulness of Training |
|---|---|---|
| Career growth | Pearson's correlation | 0.593 ** |
| | Sig. (2-tailed) | 0.000 |
| | N | 150 |

** Level of significance 0.01.

### 4.3. H3

The attitude of the media company toward the additional education of employees was examined based on five statements, stated in the questionnaire:

S14: The media company where I am employed provides regular and adequate training for journalists and editors in order for them to improve their skills.

S15: My superior shows interest in my personal and professional development.

S16: There is a training plan for each job position.

S17: The choice of training included in the training plan is in accordance with the skills I need to perform my job.

S18: The choice of training that is included in the training plan for my workplace could enable me to improve my personal professional ability.

Additionally, with the help of correlation analysis, it was examined whether the attitude of the media company toward the additional education of employees is related to career growth.

Table A4 shows the results of the correlation between media companies' attitudes toward training and career growth. In the table, it can be determined that there are statistically significant correlations between all of the examined statements. All obtained correlations are positive, which can be interpreted so that with the increase in the media company's positive attitude toward training, the positive attitude toward career growth also increases.

Table 5 shows that there has been a statistically significant, strong, positive correlation between career growth and the organizational support for employees' training.

**Table 5.** Organizational support for training.

| | | Organizational Support for Training |
|---|---|---|
| Career growth | Pearson's correlation | 0.777 ** |
| | Sig. (2-tailed) | 0.000 |
| | N | 150 |

** Level of significance 0.01.

In order to examine the influence of the subjective assessment of the need for training, the usefulness of training, and the organizational support for training on career growth, regression analysis was applied. These predictor variables in the obtained model (subjective assessment of the need for training, subjective assessment of the usefulness of training, and the organizational support for training) explain a 64% variation in the dependent variable (career growth). This model is statistically significant. As statistically significant predictors, the subjective assessment of the usefulness of training and organizational support for training stand out. Both predictors are positive, so it can be concluded that as the subjective assessment of the usefulness of training increases, career growth increases by 0.181. Furthermore, as the organizational support for training increases, career progress growth increases by 0.651. At the same time, organizational support for training has a stronger impact on career growth (Table 6). Based on these results, it can be concluded that hypothesis H3 has been confirmed.

**Table 6.** The influence of the training elements on career growth.

| Model | | Unstandardized Coefficients | | Standardized Coefficients | t | Sig. |
|---|---|---|---|---|---|---|
| | | **R = 0.797, R Square = 0.635** | | | | |
| | | **F = 84.758, Sig. = 0.001** | | | | |
| | | **B** | **Std. Error** | **Beta** | | |
| 1 | (Constant) | 0.214 | 0.285 | | 0.753 | 0.453 |
| | The subjective assessment of the need for training | 0.094 | 0.066 | 0.076 | 1.419 | 0.158 |
| | The subjective assessment of the usefulness of training | 0.191 | 0.068 | 0.181 | 2.820 | 0.005 |
| | Organizational support for training | 0.564 | 0.054 | 0.651 | 10.482 | 0.000 |

### 4.4. H4

To examine hypothesis H4 and to detect which socio-demographic characteristics interfere with the attitude of the employees toward training regarding their career growth, a linear regression model with a stepwise selection method was used.

To measures the attitude of the employees toward training regarding their career growth, a numerical variable was generated as independent on a scale of 1 to 5 as an average of variables' subjective assessment of the need for training. The higher the value of the newly generated variable, the more the employee is interested in finding opportunities for learning and additional training, proactive in seeking professional development, and ready to learn more to achieve professional goals. This variable was used as the dependent variable in the linear regression model.

As independent variables, all available socio-demographic characteristics of employees were considered (gender, age, belonging to a media company, prevalence of the media in which the respondents are employed, and job position). Categorical variables are entered into the model as dummy variables for each category. The selected variables in the final model are dummy variables for radio companies and dummy variables for the job position of the collaborator. The ANOVA test confirms that the considered regression model is adequate (Table 7).

**Table 7.** Linear regression: ANOVA.

| Model | Sum of Squares | df | Mean Square | F | Sig. |
|---|---|---|---|---|---|
| | | **ANOVA** | | | |
| Regression | 104.848 | 2 | 52.424 | 13.815 | 0.000 |
| Residual | 557.826 | 147 | 3.795 | | |
| Total | 662.673 | 149 | | | |

Estimated model coefficients are given in Table 8. People employed on the radio are less interested in career development than people from other media industries. People in the collaborator position are more interested in career development than people in other positions.

The results for hypothesis H4 point toward the fact that general demographic characteristics are not a deciding factor in career growth, but education is. This leads us to believe that the significance lies precisely in what hypotheses H1, H2, and H3 have stipulated: that professional training has an impact on the career growth of the employees and this is what company management needs to direct their attention toward.

**Table 8.** Linear regression: model coefficients.

| Model | Unstandardized Coefficients | | T | Sig. |
|---|---|---|---|---|
| | B | Std. Error | | |
| (Constant) | 12.698 | 0.186 | 68.146 | 0.000 |
| Company = radio | −1.696 | 0.442 | −3.836 | 0.000 |
| Job position = collaborator | 1.590 | 0.421 | 3.779 | 0.000 |

***R2 = 15%.

There is no statistically significant difference in the ratings of all three claims of the subjective assessment of the need for training in relation to gender (S8: Mann–Whitney $U = 2634$, $p = 0.467$; S9: Mann–Whitney $U = 2458$, $p = 0.162$; S10: Mann–Whitney $U = 2679.5$, $p = 0.543$) or age (S8: chi-square = 5.792, $p = 0.215$; S9: chi-square = 6.168, $p = 0.187$; S10: chi-square = 4.488, $p = 0.344$).

There is a statistically significant difference in the subjective assessment of the need for training regarding the media company where the respondents are employed (S8: chi-square = 4.890, $p = 0.430$; S9: chi-square = 13.856, $p = 0.017$; S10: chi- square = 32.783, $p = 0.000$). The most proactive were the employees of news agencies, while the least proactive respondents were the employees of radio stations. The most eager for additional training were employees in media production, while the least eager were the employees of radio stations (Table A5). It was found that there is no statistically significant difference in the organization of regular training (S14: chi-square = 10.342, $p = 0.066$) and the interest of their superiors (S15: chi-square = 10.342, $p = 0.301$), as well as the training plan (S16: chi-square = 10.342, $p = 0.066$) regarding the company where the respondents are employed (Tables A6 and A7).

The statistical significance of training attendance was also examined in the case of the following categories: gender, age, belonging to a media company, prevalence of the media in which the respondents are employed, and job position (please see Appendix B—Figures A1–A5):

- There is no statistically significant difference in training attendance in relation to gender (chi-square = 5.186, $p = 0.075$) (Figure A1).
- There is a statistically significant difference in training attendance in relation to age (chi-square = 26.2609, $p = 0.001$) at the 1% level of statistical significance. Members of the age category between 41 and 50 attended the most training sessions during their careers (Figure A2).
- There is no statistically significant difference in attending training in relation to the media company (chi-square = 6.6216, $p= 0.761$) (Figure A3).
- There is no statistically significant difference in training attendance in relation to the media distribution area in which the respondents are employed (chi-square = 8.950, $p = 0.062$) (Figure A4).
- There is a statistically significant difference in training attendance in relation to the job position where the respondents are employed (chi-square = 29.051, $p = 0.000$) at the level of statistical significance of 1%. Employees in the position of editors and journalists attended most additional training during their careers (Figure A5).

## 5. Discussion

The personal attitude of employees toward professional training, as well as the attitude of the organization toward employee training and education programs, are significant from the aspect of career progress, in the sense of better performance at work, moving up the hierarchical ladder, changes in earnings, etc., all in the wider context of motivation of employees and maximum efficiency in performing the tasks that are set before them with the ultimate goal of satisfying the employer, in this case, a media company.

The results of this research confirmed the research hypotheses, that is, employees in media companies believe that additional training has an impact on the development of

their careers, as well as that attending additional training has a positive effect on their career growth. Both the personal attitudes of the employees and the organization's positive attitude toward professional training are highly correlated with the career growth of employees in media companies.

It can be concluded that it is necessary to invest in the training of employees in order to achieve their sustainable careers and, therefore, the sustainability of the operations of media companies. Bozionelos, Lin, and Lee [8] indicated that training enables individuals to update their existing knowledge but also to acquire new "knowledge, skills, and work-related values and attitudes, with which to discover new opportunities and sustain their suitability and value for current and future employers" (p. 4). Increasing the existing knowledge leads to two positive outcomes: a satisfied employer due to the efficiency of the employees' work and satisfied employees because they perform their jobs with ease and are motivated due to personal progress.

The results of research conducted by Slavic et al. [16] indicate that companies from former socialist countries (Croatia, Serbia, Slovakia, Slovenia, and Hungary) carry out a systematic evaluation of the effectiveness of training, as well as that they mostly use on-the-job training and mentoring. Moreover, in countries such as Austria, Germany, Denmark, and Sweden, there are institutionalized training programs for journalists. Some of the authors [41–44] point out that, for example, in Switzerland, Austria, and Portugal, the development of journalistic skills primarily depends on the initiative of the journalists themselves, while some of them, such as those from Hong Kong, are forced to find additional training abroad [26]. In this regard, the authors of this paper conducted research on the topic of training employees in media companies in Serbia and confirmed the mutual connection between training and career growth of those employed by media companies.

The results of this research should be seen as a recommendation for management in media companies in the field of personal development and additional education of employees, in order to expand their knowledge, skills, and abilities to perform the work they are engaged in, with the ultimate goal of increasing the efficiency and productivity in the workplace. In this way, the managers of media companies would be introduced to the topic of the sustainability of their business in modern conditions.

Human resources play a significant role in guiding companies toward achieving competitive advantages. Professional training and greater competencies of employees are certainly one of the factors of competitive advantage. For this reason, it is necessary to understand the importance of training organized with the aim of improving the knowledge, skills, and motivation of people working in small and medium-sized enterprises [36], which certainly includes numerous media houses.

In the long term, education and learning are important components of the EU's strategy for smart, sustainable, and inclusive growth. Job-related learning and continuous professional development with the practice of training and education play a key role in this regard [45]. Businesses in Serbia, including media companies, strive to align their operations with EU trends, which certainly also applies to trends in terms of training, i.e., professional development and advancement of their employees. This will certainly contribute to the success and efficiency of the company's work and the greater satisfaction of its employees. The results have shown that the amount of monthly income and satisfaction with the employer both have a significant effect on the attitude of employees toward their growth in the company [46].

### 5.1. Theoretical and Practical Implications

This paper has identified clear practical implications since the results of this research may serve as a clear guideline for human resource managers in media companies in approaching the organization of employee training and development. This would eventually have a positive impact on the career growth of the employees but also the better placement of the media companies on the market. This is of utmost importance since the employees see career growth as a motivator for work. Moreover, a satisfied employee is more likely to

stay in the company and, at the same time, there is a mutual benefit, both for the employer (media house) and for the employee. For this reason, the results of this research can find great applicability in practice, which reflects in the confirmation of the significance of organizing and conducting professional training, which is often neglected by managers in media companies. The employees have developed awareness of the importance of professional development; however, they also need organizational support. Career growth through continuous development is not significant only from the standpoint of media companies, but also in all other spheres of business. The implications for society reflected the fact that media companies and their management must influence the employees' status in the companies and provide adequate and professional working conditions.

The results of this research should be seen as a recommendation for the management of media companies regarding personal development and additional education of employees. Company management should place significance on the professional training of their employees in order to expand their knowledge, skills, and abilities to perform the work they are engaged in, with the ultimate goal of increasing efficiency and productivity in the workplace and compliance with the standards of the European Union.

The theoretical contribution of this work may be seen as an upgrade of the existing research [2,21] which highlighted the significance of career growth and the professional development of the employees, as well as the mutual dependence on employee training and education and their career advancement. This study fills the gap between theory and practice since the obtained results point toward the significance that professional training has for employees in their career development. This was also addressed by Bozionelos et al. [8], who claim that investing in employee training is a road toward sustainable careers and that professional training organized by companies leads to the sustainability of human resources, that is, their employees.

It is very important to examine how frequent professional training for media professionals is given, because the involvement of employees in such training shows a better organizational performance of those media professionals who attended the training [47]. Our work is a complement to such research because career progress can be reflected in organizational performance [48]. Media professionals are employees whose career progress should be accompanied by training, which enables them to use new business methods, techniques, and knowledge and reduce the possible shortcomings in their work [49].

Therefore, the contribution of the work is reflected in the observation of the subjective factors affecting the development of employees in the media sector, which is still burdened by an unfavorable status in society [50], as well as economic and political pressures that slow down its development.

### 5.2. Limitations

The main limitations of this study are related to the size of the sample and the instrument used in the research. A bit over half of the active members of only one journalist's association (The Journalists Association of Vojvodina) participated in the study, and even though these members include media professionals employed by media companies all over Serbia, there are many more such associations in this country. Hence, it would be beneficial to widen the sample and conduct this research among the members of other associations (e.g., The Independent Association of Journalists of Serbia, The Association of journalists of Serbia, etc.). The research sample for future studies can be expanded to many other fields of business as well, such as manufacturers and service providers.

The instrument used poses the second limitation of this study since only those elements of the scale which investigate career growth, and those parts of the subscales which measure the attitudes toward training (individual and organizational), as well as the applicability of the training were used. In future research, the whole scale for assessing career growth can be used, as well as the whole scale for training assessment. It is possible also to look into the particular types of training that would reflect positively on the employees' career growth.

*5.3. Suggestions for Future Research*

The proposal for future research would be to conduct the same research in a wider geographical area;for example, in the Balkan region, in order to gain insight into the attitudes of employees toward professional training, but also into the impact professional training has on the further development and sustainability of the careers of employees in media companies. Furthermore, it would be desirable to repeat the research after a certain period of time, in the same territory, but with a larger sample. In this way, the set hypotheses would be reaffirmed, which would be of great importance for undertaking specific activities that could be implemented in the human resources sectors in media companies, with the aim of further improving the work of employees, based on expertise, competitiveness, and increased market competitiveness. Another suggestion for future research is to conduct a survey among the managers of media companies, in order to determine their attitudes regarding the proposed topic of training and workplace training.

Certainly, as a recommendation for future research, it is necessary to evaluate the effectiveness of conducted training in media companies and check whether there are on-the-job training and mentoring sessions.

**Author Contributions:** Conceptualization, B.S. and I.Š.; methodology, B.S., I.Š. and S.K.; distribution of the questionnaire to respondents and checking the data B.S., L.D. and S.K.; conducting the statistical analysis, B.S. and L.D.; writing—original draft preparation, B.S., I.Š. and L.D.; writing—review and editing, S.K., B.S., I.Š. and L.D.; visualization, B.S. and S.K.; supervision, B.S. and I.Š. All authors contributed to the interpretation of the data and editing of the manuscript. All authors have read and agreed to the published version of the manuscript.

**Funding:** This research received no external funding.

**Institutional Review Board Statement:** Not applicable.

**Informed Consent Statement:** Informed consent was obtained from all subjects involved in the study.

**Data Availability Statement:** The data that support the findings of this study are available from the corresponding author upon reasonable request.

**Conflicts of Interest:** The authors declare no conflict of interest.

## Appendix A

The attitude of media professionals toward additional training and professional development.

**Questionnaire**

I　　Demographic characteristics of research participants

1.　What is your gender?
    (a)　Male
    (b)　Female

2.　What is your age?
    (a)　Less than 20
    (b)　20–30
    (c)　31–40
    (d)　41–50
    (e)　More than 50

3.　Which media company do you work for?
    (a)　On TV
    (b)　Radio
    (c)　In a newspaper company
    (d)　Internet portal
    (e)　Newspaper agency

      (f)      Media production

4.     In which area of the media are you employed?

      (a)      National media
      (b)      Regional media
      (c)      Local media

5.     What is your job position?

      (a)      Journalist
      (b)      Editor
      (c)      Collaborator
      (d)      Other

6.     How many years of work experience do you have in the media industry? (Enter the number of years)

II     The attitude of media professionals towards additional training and advancement.

7.     Please evaluate the following statements in accordance with your attitude towards additional education:

(Scale: 1-strongly disagree, 2-disagree, 3-undecided, 4-partially agree, 5-completely agree; write X in the selected field)

| Statements: | 1 | 2 | 3 | 4 | 5 |
|---|---|---|---|---|---|
| I always find opportunities for learning and additional training. | | | | | |
| I am proactive in looking for ways to improve professionally. | | | | | |
| I am ready to learn more in order to achieve my professional goals. | | | | | |

8.     Have you attended any training organized with the aim of additional business education during your career?

      (a)      Yes, more than once.
      (b)      Yes, once.
      (c)      No, I have never attended professional training.

9.     Please rate the following statements according to the degree of satisfaction with the trainings you have attended during your career:

(Scale: 1-strongly disagree, 2-disagree, 3-undecided, 4-partially agree, 5-completely agree; write X in the selected field)

| Statements: | 1 | 2 | 3 | 4 | 5 |
|---|---|---|---|---|---|
| The knowledge I acquired during my journalistic education significantly contributed to the quality of the work I perform. | | | | | |
| I believe that the knowledge gained during training in my field is very applicable in my work. | | | | | |
| I am very satisfied with the frequency of training organized by the media company where I am employed. | | | | | |

III     The attitude of the media company towards additional training of media professionals

10.     Please evaluate the following statements regarding the implementation of the training organized by the media company where you are employed:

(Scale: 1-strongly disagree, 2-disagree, 3-undecided, 4-partially agree, 5-completely agree; write X in the selected field)

| Statements: | 1 | 2 | 3 | 4 | 5 |
|---|---|---|---|---|---|
| The media company where I am employed provides regular and adequate training for journalists and editors in order to improve their skills. | | | | | |
| My superior shows interest in my personal and professional development. | | | | | |
| There is a training plan for each job position. | | | | | |
| The choice of training included in the training plan is in accordance with the skills I need to perform my job. | | | | | |
| The choice of training that is included in the training plan for my workplace could enable me to improve my personal professional ability. | | | | | |

11. Please estimate the frequency of training organized by the media company where you are employed:

(Scale: 1-often, 2-rarely, 3-never; write X in the selected field)

| Statements: | 1 | 2 | 3 |
|---|---|---|---|
| The company organizes "in house" training held by media professionals employed from the same company. | | | |
| The company organizes "in house" training (workplace training) held by media professionals from other media organizations. | | | |
| The company provides funds for external training. | | | |

IV The attitude of media professionals towards personal career growth

12. Please evaluate the following statements in accordance with personal professional development:

(Scale: 1-strongly disagree, 2-disagree, 3-undecided, 4-partially agree, 5-completely agree; write X in the selected field)

| Statements: | 1 | 2 | 3 | 4 | 5 |
|---|---|---|---|---|---|
| If I attend trainings according to the set training plan, I will increase the chances of reaching my career goals. | | | | | |
| My supervisor encourages me to constantly acquire new knowledge and skills related to the current job. | | | | | |
| The trainings that I would attend, which are foreseen by the training plan, can enable me to progress in the organizational ladder in the next year. | | | | | |
| My current job allows me to continuously improve my professional skills. | | | | | |
| The probability of advancement in the media company where I am employed is high. | | | | | |
| Compared to my colleagues, I was promoted faster. | | | | | |
| The knowledge and skills I have acquired through training during my career have contributed to my progress, which is reflected in an increase in salary. | | | | | |

## Appendix B

**Table A1.** Attending training sessions—individual statements.

|  | Yes, More than One | Yes, One | No |
|---|---|---|---|
| S1: If I attend training according to the set training plan, I will increase the chances of reaching my career goals. | 3.52 | 2.90 | 2.63 |
| S2: My supervisor encourages me to constantly acquire new knowledge and skills related to the current job. | 3.20 | 2.59 | 2.81 |
| S3: The training that I would attend, which are foreseen by the training plan, can enable me to progress in the organizational ladder in the next year. | 3.15 | 2.93 | 2.22 |
| S4: My current job allows me to continuously improve my professional skills | 3.61 | 3.14 | 3.15 |
| S5: The probability of advancement in the media company where I am employed is high | 2.76 | 3.03 | 2.04 |
| S6: Compared to my colleagues, I was promoted faster | 2.73 | 3.10 | 2.22 |
| S7: The knowledge and skills that I have acquired through training during my career have contributed to my progress, which is reflected in the increase in salary. | 2.46 | 2.62 | 2.07 |

**Table A2.** The subjective assessment of the need for training based on demographic characteristics—individual items.

|  | S8: I Always Find Opportunities for Learning and Additional Training | | S9: I Am Proactive in Looking for Ways to Improve Professionally | | S10: I Am Ready to Learn More in Order to Achieve My Professional Goals | |
|---|---|---|---|---|---|---|
|  | Correlation Coef. | $p$ | Correlation Coef. | $p$ | Correlation Coef. | $p$ |
| S1: If I attend training according to the set training plan, I will increase the chances of reaching my career goals. | 0.311 ** | 0.000 | 0.286 ** | 0.000 | 0.229 ** | 0.005 |
| S2: My supervisor encourages me to constantly acquire new knowledge and skills related to the current job. | 0.206 * | 0.011 | 0.234 ** | 0.004 | −0.074 | 0.365 |
| S3: The training that I would attend, which are foreseen by the training plan, can enable me to progress in the organizational ladder in the next year. | 0.300 ** | 0.000 | 0.255 ** | 0.001 | 0.275 ** | 0.001 |

**Table A2.** *Cont.*

| | S8: I Always Find Opportunities for Learning and Additional Training | | S9: I Am Proactive in Looking for Ways to Improve Professionally | | S10: I Am Ready to Learn More in Order to Achieve My Professional Goals | |
|---|---|---|---|---|---|---|
| | Correlation Coef. | *p* | Correlation Coef. | *p* | Correlation Coef. | *p* |
| S4: My current job allows me to continuously improve my professional skills | 0.150 | 0.067 | 0.131 | 0.120 | −0.078 | 0.343 |
| S5: The probability of advancement in the media company where I am employed is high | 0.240 ** | 0.003 | 0.194 * | 0.018 | 0.036 | 0.659 |
| S6: Compared to my colleagues, I was promoted faster | 0.111 | 0.175 | 0.109 | 0.185 | 0.220 ** | 0.007 |
| S7: The knowledge and skills that I have acquired through training during my career have contributed to my progress, which is reflected in the increase in salary. | 0.240 ** | 0.003 | 0.166 * | 0.043 | 0.094 | 0.249 |

\* Level of significance 0.05. \*\* Level of significance 0.01.

**Table A3.** The employees' subjective assessment of the usefulness of training—individual items.

| | S11: The Knowledge I Acquired during My Journalism Education Significantly Contributed to the Quality of the Work I Do | | S12: I Believe That the Knowledge Gained at the Training in My Field Is Very Applicable in My Work | | S13: I Am Very Satisfied with the Frequency of Training Organized by the Media Company Where I Am Employed | |
|---|---|---|---|---|---|---|
| | Correlation Coef. | *p* | Correlation Coef. | *p* | Correlation Coef. | *p* |
| S1: If I attend training according to the set training plan, I will increase the chances of reaching my career goals. | 0.502 ** | 0.000 | 0.466 ** | 0.000 | 0.411 ** | 0.000 |
| S2: My supervisor encourages me to constantly acquire new knowledge and skills related to the current job. | 0.293 ** | 0.000 | 0.188 * | 0.021 | 0.461 ** | 0.000 |

<div align="center"><b>Table A3.</b> <i>Cont.</i></div>

| | S11: The Knowledge I Acquired during My Journalism Education Significantly Contributed to the Quality of the Work I Do | | S12: I Believe That the Knowledge Gained at the Training in My Field Is Very Applicable in My Work | | S13: I Am Very Satisfied with the Frequency of Training Organized by the Media Company Where I Am Employed | |
|---|---|---|---|---|---|---|
| | Correlation Coef. | *p* | Correlation Coef. | *p* | Correlation Coef. | *p* |
| S3: The training that I would attend, which are foreseen by the training plan, can enable me to progress in the organizational ladder in the next year. | 0.356 ** | 0.000 | 0.377 ** | 0.000 | 0.534 ** | 0.000 |
| S4: My current job allows me to continuously improve my professional skills | 0.214 ** | 0.009 | 0.177 * | 0.030 | 0.303 ** | 0.000 |
| S5: The probability of advancement in the media company where I am employed is high | 0.143 | 0.080 | 0.157 | 0.055 | 0.483 ** | 0.000 |
| S6: Compared to my colleagues, I was promoted faster | 0.100 | 0.223 | 0.170 * | 0.038 | 0.488 ** | 0.000 |
| S7: The knowledge and skills that I have acquired through training during my career have contributed to my progress, which is reflected in the increase in salary. | 0.296 ** | 0.000 | 0.215 ** | 0.008 | 0.524 ** | 0.000 |

* Level of significance 0.05. ** Level of significance 0.01.

<div align="center"><b>Table A4.</b> The media company's attitude toward training—organizational support for training—individual items.</div>

| | S14: The Media Company Where I Am Employed Provides Regular and Adequate Training for Journalists and Editors in Order to Improve Their Skills. | | S15: The Superior Is Interested in My Personal and Professional Development | | S16: There Is a Training Plan for Each Job Position | | S17: The Choice of Training Included in the Training Plan Is in Accordance with the Skills I Need to Perform My Job | | S18: The Choice of Training That Is Included in the Training Plan for My Workplace Could Enable Me to Improve My Personal Professional Ability | |
|---|---|---|---|---|---|---|---|---|---|---|
| | r | p | r | p | r | p | r | p | r | p |
| S1 | 0.551 ** | 0.000 | 0.445 ** | 0.000 | 0.355 ** | 0.000 | 0.483 ** | 0.000 | 0.570 ** | 0.000 |
| S2 | 0.570 ** | 0.000 | 0.707 ** | 0.000 | 0.554 ** | 0.000 | 0.535 ** | 0.000 | 0.616 ** | 0.000 |
| S3 | 0.456 ** | 0.000 | 0.448 ** | 0.000 | 0.507 ** | 0.000 | 0.418 ** | 0.000 | 0.528 ** | 0.000 |
| S4 | 0.430 ** | 0.000 | 0.516 ** | 0.000 | 0.492 ** | 0.000 | 0.570 ** | 0.000 | 0.632 ** | 0.000 |
| S5 | 0.433 ** | 0.000 | 0.491 ** | 0.000 | 0.554 ** | 0.000 | 0.431 ** | 0.000 | 0.488 ** | 0.000 |
| S6 | 0.447 ** | 0.000 | 0.246 ** | 0.002 | 0.267 ** | 0.001 | 0.212 ** | 0.009 | 0.303 ** | 0.000 |
| S7 | 0.488 ** | 0.000 | 0.460 ** | 0.000 | 0.459 ** | 0.000 | 0.386 ** | 0.000 | 0.417 ** | 0.000 |

** Level of significance 0.01.

**Table A5.** The subjective assessment of the need for training based on demographic characteristics.

| | S8: I Always Find Opportunities for Learning and Additional Training | S9: I Am Proactive in Seeking Ways for Professional Development | S10: I Am Ready to Learn More in Order to Achieve My Professional Goals |
|---|---|---|---|
| Men | 4.18 | 3.84 | 4.57 |
| Women | 4.24 | 3.99 | 4.62 |
| Total | 4.21 | 3.91 | 4.59 |
| <20 | 5.00 | 5.00 | 5.00 |
| 20–30 | 4.31 | 4.06 | 4.69 |
| 31–40 | 4.33 | 3.97 | 4.70 |
| 41–50 | 4.11 | 3.63 | 4.39 |
| Over 50 | 3.86 | 4.05 | 4.50 |
| Total | 4.21 | 3.91 | 4.59 |
| On TV | 4.25 | 3.95 | 4.76 |
| On radio | 3.91 | 3.48 | 3.96 |
| In a newspaper company | 4.14 | 3.62 | 4.57 |
| On an internet portal | 4.31 | 4.25 | 4.67 |
| Newspaper agency | 4.40 | 4.40 | 4.40 |
| Media production | 4.30 | 3.90 | 5.00 |
| Total | 4.21 | 3.91 | 4.59 |
| National media | 4.27 | 3.86 | 4.61 |
| Regional media | 4.30 | 3.98 | 4.52 |
| Local media | 4.06 | 3.90 | 4.68 |
| Total | 4.21 | 4.00 | 4.59 |
| Journalist | 4.08 | 3.73 | 4.60 |
| Editor | 4.22 | 3.76 | 4.56 |
| Collaborator | 4.58 | 4.54 | 4.85 |
| Other | 4.06 | 4.06 | 4.25 |
| Total | 4.21 | 3.91 | 4.59 |

**Table A6.** The employees' subjective assessment of the usefulness of training based on demographic characteristics.

| | S11: The Knowledge I Acquired during My Journalism Education Significantly Contributed to the Quality of the Work I Do | S12: I Believe That the Knowledge Gained at the Training in My Field Is Very Applicable in My Work | S13: I Am Very Satisfied with the Frequency of Training Organized by the Media Company Where I Am Employed |
|---|---|---|---|
| On TV | 3.80 | 3.93 | 2.82 |
| On radio | 3.61 | 3.61 | 2.91 |
| In a newspaper company | 4.00 | 4.14 | 1.81 |
| On an internet portal | 4.00 | 3.94 | 3.28 |
| Newspaper agency | 3.00 | 4.60 | 3.40 |
| Media production | 3.70 | 3.80 | 2.50 |
| Total | 3.81 | 3.93 | 2.80 |

**Table A6.** *Cont.*

| | S11: The Knowledge I Acquired during My Journalism Education Significantly Contributed to the Quality of the Work I Do | S12: I Believe That the Knowledge Gained at the Training in My Field Is Very Applicable in My Work | S13: I Am Very Satisfied with the Frequency of Training Organized by the Media Company Where I Am Employed |
|---|---|---|---|
| National media | 3.73 | 3.96 | 2.63 |
| Regional media | 4.00 | 4.04 | 3.12 |
| Local media | 3.71 | 3.78 | 2.65 |
| Total | 3.81 | 3.93 | 2.80 |
| Journalist | 3.59 | 3.83 | 2.49 |
| Editor | 4.27 | 4.31 | 3.07 |
| Collaborator | 3.81 | 3.77 | 3.08 |
| Other | 3.44 | 3.50 | 2.81 |
| Total | 3.81 | 3.93 | 2.80 |

**Table A7.** Organizational support for training based on demographic characteristics.

| | S14: The Media Company Where I Am Employed Provides Regular and Adequate Training for Journalists and Editors in Order to Improve Their Skills. | S15: The Superior Is Interested in My Personal and Professional Development | S16: There Is a Training Plan for Each Job Position | S17: The Choice of Training Included in the Training Plan Is in Accordance with the Skills I Need to Perform My Job | S18: The Choice of Training That Is Included in the Training Plan for My Workplace Could Enable Me to Improve My Personal Professional Ability |
|---|---|---|---|---|---|
| On TV | 2.69 | 3.15 | 2.31 | 3.24 | 3.35 |
| On radio | 3.04 | 3.04 | 2.30 | 3.09 | 2.87 |
| In a newspaper company | 2.00 | 2.48 | 2.24 | 2.81 | 2.86 |
| On an internet portal | 2.86 | 3.28 | 2.69 | 3.03 | 3.31 |
| Newspaper agency | 2.40 | 2.80 | 3.80 | 3.40 | 4.00 |
| Media production | 2.90 | 3.50 | 2.30 | 3.30 | 3.10 |
| Total | 2.69 | 3.08 | 2.44 | 3.11 | 3.20 |
| National media | 2.57 | 3.20 | 2.39 | 2.88 | 3.16 |
| Regional media | 3.02 | 2.88 | 2.38 | 3.14 | 3.10 |
| Local media | 2.49 | 3.16 | 2.55 | 3.31 | 3.33 |
| Total | 2.69 | 3.08 | 2.44 | 3.11 | 3.20 |
| Journalist | 2.35 | 2.62 | 2.21 | 2.78 | 2.97 |
| Editor | 2.73 | 3.24 | 2.42 | 3.31 | 3.24 |
| Collaborator | 3.15 | 3.46 | 2.85 | 3.58 | 3.46 |
| Other | 3.19 | 3.81 | 2.75 | 3.13 | 3.56 |
| Total | 2.69 | 3.08 | 2.44 | 3.11 | 3.20 |

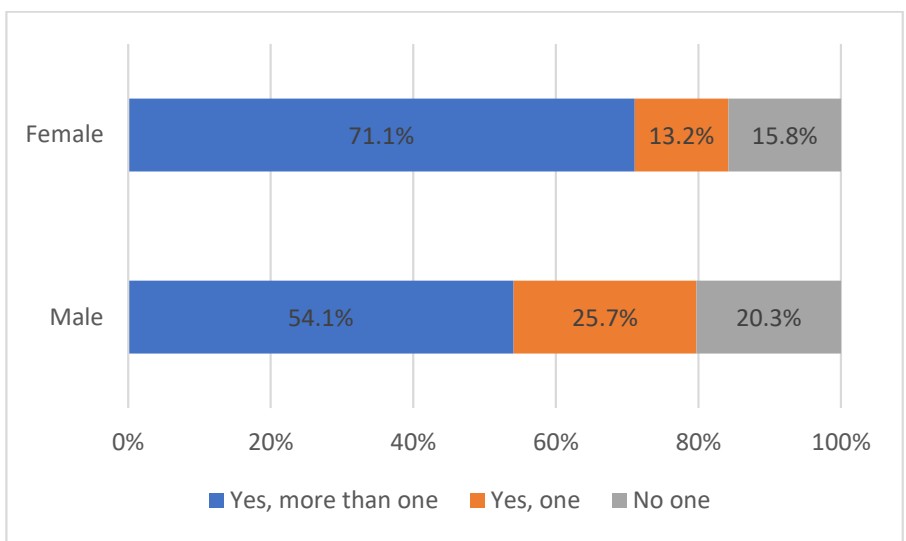

**Figure A1.** The statistical significance of training attendance depending on gender.

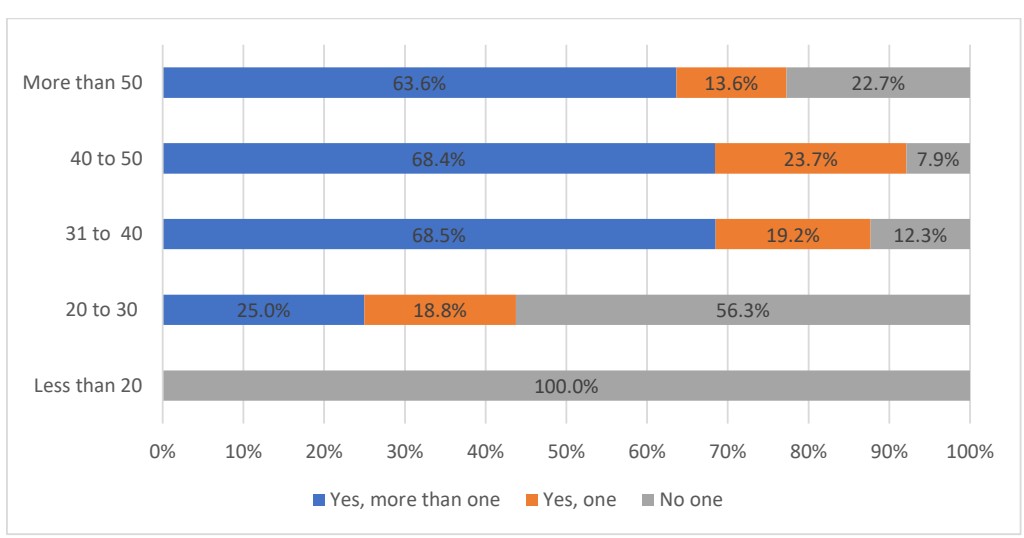

**Figure A2.** The statistical significance of training attendance depending on the age of respondents.

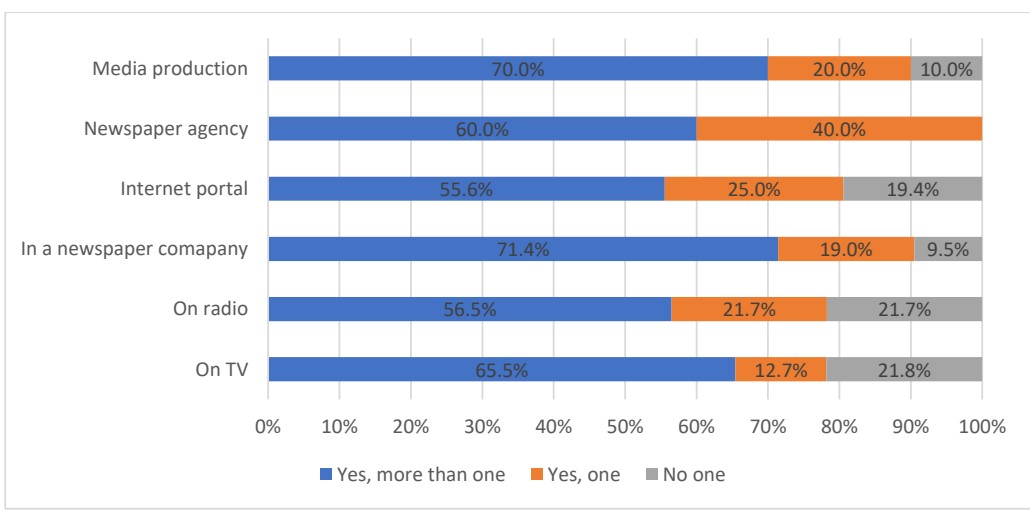

**Figure A3.** The statistical significance of training attendance depending on belonging to a media company.

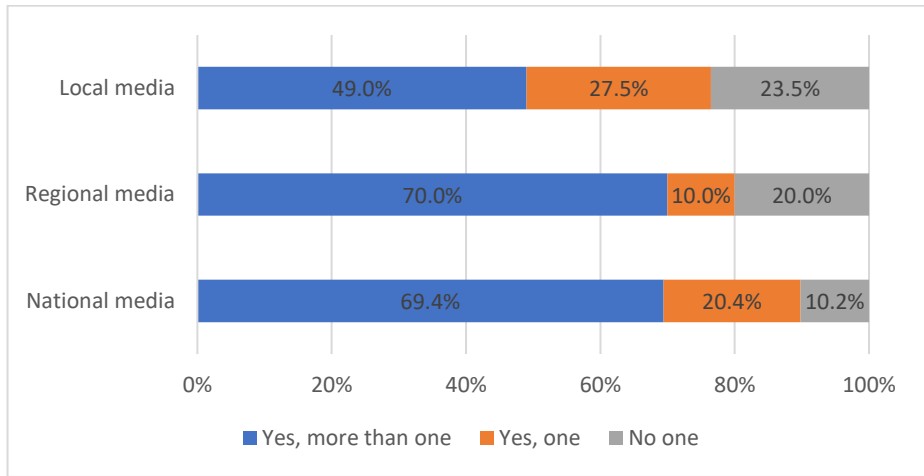

**Figure A4.** The statistical significance of training attendance depending on the prevalence of the media in which the respondents are employed.

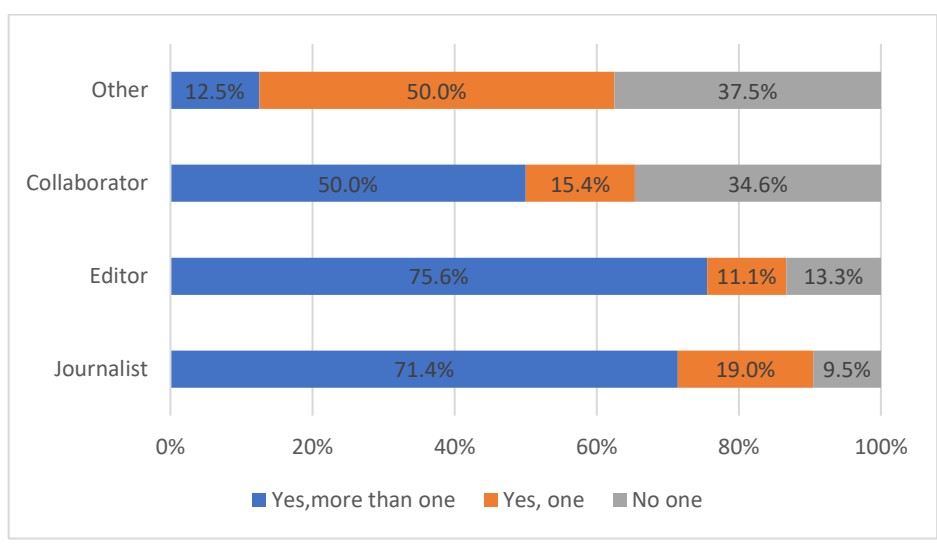

**Figure A5.** The statistical significance of training attendance depending on the job position.

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
