# Peer review of "Professional Training of Employees in Media Organizations in Serbia and Its Implications on Career Development"

_sustainability, doi:10.3390/su15054105_

Round 1

Reviewer 1 Report

These are my comments on the manuscript

1. The abstract did not mentioned any method applied in this study? 

2. L60, P2, "The authors of [3] especially"- Just state the authors name.

3. L102-104 " In order for an individual to get and keep a job in a changing labor market, he must increase his needs for career competencies that can help him manage his own career [9]"

- Please rephrase the sentence without referring to "he" and "his"

4. L129 "Let's add another important dimension, that concerns the employee himself, and that is his career development." - Rephrase this sentence to be more formal. And short form is not suitable for academic writing e.g. Let's

5. L175 "We are talking about the  sustainability of human resources in organizations, and this certainly applies to 176 employees in media companies as well" - please refrain from using first name, e.g. "I", "we" and "us".

6. L215 "Accordingly, we cannot bypass the increasingly intensive development of robotic or automated journalism, respectively software that is programmed to write a simpler news report and relieve journalists of their daily routine tasks."- same issue of using first name.

7. The hypotheses should be independent from method. It should be before, Suggest to create a section of hypothesis development.

8. L507 " By increasing the existing  knowledge, we arrive at two results"

- It should be conclusion rather than result

9. There should be sections of implications, future works and limitations. Even though implications are discussed, author should create a new section for it, making it easy for readers. There no limitations discussed.

10. The manuscript requires English editing and proofreading

Reviewer 2 Report

Additional Questions:

1.      Originality:  Does the paper contain new and significant information adequate to justify publication?: Yes. he authors identified an interesting topic and provided significant information on the relation of Professional Training of Employees in Media Organization in Serbia and its implications on Career Development”. However, there are some issues that need clarification. More importantly, the discussion regarding the research gaps is missing!

2.      Relationship to Literature:  Does the paper demonstrate an adequate understanding of the relevant literature in the field and cite an appropriate range of literature sources?  Is any significant work ignored?: The literature review section is fine.

3.      Methodology:  Is the paper's argument built on an appropriate base of theory, concepts, or other ideas?  Has the research or equivalent intellectual work on which the paper is based been well designed?  Are the methods employed appropriate?: The paper is well structured and follows the standards.

4. Results:  Are results presented clearly and analysed appropriately?  Do the conclusions adequately tie together the other elements of the paper?: The results section is fine.

5. Implications for research, practice and/or society:  Does the paper identify clearly any implications for research, practice and/or society?  Does the paper bridge the gap between theory and practice? How can the research be used in practice (economic and commercial impact), in teaching, to influence public policy, in research (contributing to the body of knowledge)?  What is the impact upon society (influencing public attitudes, affecting quality of life)?  Are these implications consistent with the findings and conclusions of the paper?: The implications part missing.

6. Quality of Communication:  Does the paper clearly express its case, measured against the technical language of the field and the expected knowledge of the journal's readership?  Has attention been paid to the clarity of expression and readability, such as sentence structure, jargon use, acronyms, etc.: A professional review of the language is strongly suggested because several parts of the text are unclear.

Reviewer 3 Report

The structure is not complete and the significance is not very valuable.

Reviewer 4 Report

Thank you for submitting the manuscript id sustainability-2160413 entitled “Professional Training of Employees in Media Organization in Serbia and its implications on Career Development.” Please see my comments in pdf

Good luck.

Round 2

Reviewer 1 Report

Thank you for providing a revision of this paper. There are two issues still that need to be highlighted:

1. Hypothesis development. Each of the hypothesis need to be developed. Please refer to good papers out there that did the hypothesis development correctly. It has to be individually not listed as like the current version.

2. 5.1. Limitations, theoretical and practical implications of this paper and proposals for future. Limitations and suggestion for future work can be together but not with implications. Please separate this.

Reviewer 3 Report

congratuations! your work is now much better, before it is published it is better to explain the contribution of your study to the existing literature.

Reviewer 4 Report

Thank you for submitting the revised manuscript id sustainability-2160413 entitled “Professional Training of Employees in Media Organization in Serbia and its implications on Career Development.” The author(s) have addressed most of my queries. However, some revisions (See pdf) need to be done.

Good luck
